



# Development and performance optimization of a parallel computing infrastructure for an unstructured-mesh modelling framework

Zhuang Liu[1,3], Yi Zhang[4], Xiaomeng Huang[1,2,3], Jian Li[4], Dong Wang[1,3], Mingqing Wang[1,3], and Xing Huang[1,3]

[1] Ministry of Education Key Laboratory for Earth System Modeling, and Department of Earth System Science, Tsinghua University, Beijing, 100084, China

[2] Laboratory for Regional Oceanography and Numerical Modeling, Qingdao National Laboratory for Marine Science and Technology, Qingdao, 266237, China

[3] National Supercomputing Center in Wuxi, Wuxi, 214071, China

[4] State Key Laboratory of Severe Weather, Chinese Academy of Meteorological Sciences, China Meteorological Administration, Beijing, 100081, China

**Correspondence:** Xiaomeng Huang (hxm@tsinghua.edu.cn)

**Abstract.** This paper describes the development and performance optimization of a
parallel computing infrastructure for an unstructured-mesh global model (GRIST;
Global-to-Regional Integrated forecast SysTem). The focus is on three major aspects
that facilitate rapid iterative development, including parallel computing, index
optimization and an efficient group I/O strategy. For parallel computing, the METIS
tool is used for the partition of the global mesh, which is flexible and convenient for
both the quasi-uniform and variable-resolution simulations. The scaling tests show
that the partition method is efficient. To improve the cache efficiency, several mesh
index reordering strategies are investigated to optimize the performance of the
indirect addressing scheme used in the stencil calculations. The numerical results
show that the indexing strategies are able to speed up the calculations, especially for
running with a small number of processes. To overcome the bottleneck of poor I/O
efficiency for the high-resolution or massively parallel simulations, a group parallel
I/O method is implemented and proven to be of high efficiency in the numerical
experiments. Altogether, these three aspects of the parallel computing toolkits are
encapsulated in a few interfaces, which can be used for general parallel modelling on
unstructured meshes.





## 1 Introduction

The global atmospheric model is an important tool for operational weather forecasting, climate prediction and research-oriented modelling. In recent years, with the continuous improvement of computing power of massively parallel computers, the global model is developed towards higher horizontal resolutions (e.g., Haarsma et al. (2016); Yu et al. (2019); Stevens et al. (2019); Dueben et al. (2020)). The unstructured grid (the semi-structured icosahedral grid and the generic Voronoi polygonal grid are considered in this study) is one of the major choices for these newly developed global models (e.g., Ullrich et al. (2017)), mainly owing to their ability to allow general computational patterns and their flexibility to switch between uniform-mesh and variable-resolution (VR) modelling.

Despite certain advantages of the unstructured meshes, several obstacles have to be overcome to achieve a practical computational efficiency. First, to support both the quasi-uniform and VR simulations, the parallel-partition strategy should be general enough and possesses a good load balance. The conventional method of dividing an icosahedral grid into 10 identical rhombi and partitioning each rhombus into blocks (e.g., MacDonald et al. (2011)) is typically not applicable. Second, the neighbours of a grid point on the unstructured meshes cannot be obtained by simple index shifting; thus, the indirect addressing scheme (MacDonald et al. (2011)) is typically used to perform the stencil calculations. This results in discontinuous memory access during model integration, which reduces the efficiency of compiler optimization and cache reuse. Although the directly addressed vertical index can be put on the innermost dimension, the computational performance in our numerical experiment is not that good, which might slightly differ from the testing conclusion of MacDonald et al. (2011) where no appreciable performance penalty for the indirect addressing scheme is observed. Third, because the mesh points distributed to each process cannot form a regular rectangular area as supported by a structured grid, the I/O operations between memory and the parallel file system are also discontinuous, posing a bottleneck for high-resolution and massively parallel computing. In short, to make scientific computing on an unstructured mesh practical, a unified and efficient approach to handle the parallel communication, computation and data I/O is an important task.



51    Recently, several works have been published for the performance optimization of

52 the unstructured-mesh models: Sinkovits et al. (2016) introduced some serial

53 optimization techniques for accelerating the dynamical core of MPAS-A[1], together

54 with a thread-level load balancing method for the atmospheric physics; Govett et al.

55 (2017) described their parallelization and optimization techniques to efficiently run

56 the Nonhydrostatic Icosahedral Model (NIM) model on CPU, GPU, and MIC

57 processors; Koldunov et al. (2019) introduced several model enhancements to

58 improve the scalability of the Finite-volumE Sea ice–Ocean Model (FESOM) for

59 large numbers of processes. On the other hand, to increase the efficiency of parallel

60 I/O, the CFIO (Climate Fast Input/Output, see Wang et al. (2013) or Huang et al.

61 (2014)) and the XIOS (XML Input/Output Server, refer to Maisonnave et al. (2017))

62 libraries applied the asynchronous computation and I/O method that uses dedicated

63 I/O processes to perform the I/O, thus overlapping the I/O phase with the computing

64 phase and shortening the entire simulation time; Dennis et al. (2011) adopted the

65 concept of defining an I/O decomposition to flexibly control the number of I/O

66 processes and rearrange the data to an I/O friendly manner, which can improve the

67 I/O throughput. In addition, the workshop "Exascale I/O for Unstructured Grids"

68 (EIUG: https://www.esiwace.eu/events/workshop-about-unstructured-grids) focused

69 on the large-scale I/O of unstructured grids, where several talks about the data formats,

70 I/O middlewares, and post-processing tools were given to deal with the I/O bottleneck

71 of the unstructured grids.

72   In this paper, we describe the development and performance optimization of a

73 parallel computing infrastructure for an unstructured-mesh global model (GRIST;

74 Global-to-Regional Integrated forecast SysTem). The GRIST framework is developed

75 based on a hierarchical structure, from a shallow water model (Zhang (2018); Wang et

76 al. (2019)) to a layer-averaged 3D dry dynamical core (Zhang et al. (2019)), and a

77 more complete moist dynamical model that supports the incorporation of model

78 physics (Zhang et al. (2020)). To facilitate rapid iterative development, we have

79 created a set of developer-friendly parallel computing toolkits to support efficient

80 establishment of numerical modelling workflow from code development to data

---

[1] The Atmospheric component of MPAS (the Model for Prediction Across Scales), refer to Skamarock et al. (2012) for more information。



evaluation. In this study, we describe three major aspects, which are tightly related to
scientific computing on an unstructured mesh. These include:
– Parallelization. We choose the METIS library (Karypis et al. (1998)) to
partition the global mesh points and design a general communication interface with an
internal collection mechanism to improve the communication efficiency. A scientific
model developer can utilize these tools without knowledge of the communication
details. The scaling test results suggest that our parallelization method is efficient.
– Index optimization. To improve the cache efficiency, we compare three
index-optimization techniques with the default unordered option. Sarje et al. (2015)
applied two space filling curve (SFC) index reordering strategies (Hilbert and Morton
curves) for the unstructured meshes and obtained 40% improvement. These two
methods         and        the        breadth-first-search         (BFS:
https://en.wikipedia.org/wiki/Breadth-first_search) strategy are considered in this
paper. We find that all the three strategies are able to accelerate the calculations, and
the BFS strategy usually generates the optimal results.
– Data I/O strategies. For improving the I/O efficiency, we have implemented a
group I/O method for the unstructured mesh. The group I/O method can combine the
small non-continuous accesses into larger continuous ones, thus increasing the I/O
granularity as well as reducing the number of I/O processes. Numerical tests show
that the group I/O method can significantly improve the I/O efficiency. A similar
strategy has also been employed by Yang et al. (2019), but for the structured grids.
Altogether, these efforts have helped model development and application and
enabled us to efficiently run GRIST at sub-10 km resolution. The rest of this paper is
organized as follows. Section 2 introduces the parallelization method. Section 3
describes the index-optimization strategies. Section 4 introduces the data I/O
optimization method. The concluding remarks are given in Section 5.

**2 Parallelization**

GRIST utilizes an unstructured icosahedral/Voronoi mesh that supports both the
quasi-uniform and VR Voronoi tessellations (Figure 1). We first define three types of
location/dimension, including the node point (the generating point with which the
primal cell is associated), the triangle point (the corner point of a Voronoi polygon





with which the dual cell is associated), and the edge point (the intersecting point of a
pair of edges that belong to a primal and dual cell, respectively). Several model
variables are located at each of the three types of mesh points. For example, the
potential temperature is located at the node point, the vorticity is located at the
triangle point, and the normal and tangent velocities are located at the edge point (see
Figure 1c in Zhang et al. (2019)). The node points can be optimized or directly
generated by the Centroidal Voronoi Tessellation (CVT) technique (e.g., Du et al.
(2003); Ringler et al. (2011)), which ensures that the generating points (node points)
are the centroids of the corresponding Voronoi cells (in the limit of the constraint).
During the model development process, two grid generators have been developed to
generate the required mesh information: one is a serial code that adopts the
STRIPACK library (Renka (1997)) to generate the Delaunay triangulations in the
iterations for optimizing the node points, and the other is a parallel code based on the
MPI-SCVT package (Jacobsen et al. (2013)). In this section, we will describe the
parallelization methods, including the mesh partition method and some techniques for
the inter-process communications.

**2.1 Mesh partition**


133       The partition of the entire global mesh can be obtained by partitioning the node

points. In practice, the METIS library is used to provide a general approach to
partition. METIS is a graph partitioning tool, which uses the input node points,
information of their neighbours and the number of partitioned groups to perform the
partition. A node point and one of its neighbours constitute two vertices of an edge in
the graph. By default, the principle of METIS is to minimize the number of edges
being cut under the constraint that the number of points assigned to each group is
roughly the same (cut-edges refer to the edges whose two vertices belong to different
groups). A smaller number of cut-edges implies less communication between groups,
and the constraint of a roughly equal number of points in each group is to ensure a
good load balance. Figure 1 illustrates a global mesh partitioned by METIS. In this
case, both the quasi-uniform and VR Voronoi cells are partitioned into ten groups.
Cells of the same colour fall in one group and will be assigned to the same process.
As a result of the partitioning principle, all processes are roughly distributed equally



for the quasi-uniform mesh, while more processes are assigned to the refinement
regions for the VR mesh.
Because the update of data on a mesh point usually requires information on its
adjacent mesh points during the model integration, each process needs the data
belonging to other processes when updating the data on its boundary mesh points (the
mesh points adjacent to mesh points of other processes). To facilitate the calculations,
three types of data areas are defined, including:
(i) Inner area: an area composed of mesh points whose data update does not
require the data from other processes;
(ii) Boundary area: an area composed of mesh points whose data update requires
the data from other processes;
(iii) Halo area: an area composed of extended mesh points in other processes for
the update of boundary data of this process.
The number of layers of the halo area can be flexibly configured. Figure 2
presents an example that uses three halo layers, while in most cases, two layers are
required (as a default). The calculation procedure for the mesh partition operates as
follows. First, we use METIS to partition the global node points, and determine three
types of areas mentioned above based on the partition and neighbourhood information
of the node points. Second, we determine the corresponding partitions of edge and
triangle points based on the partition of node points. Third, we establish the mappings
between the global and the local indices of the node, edge, and triangle points. This
completes the mesh partition.

**2.2 Communication**

Communicating with neighbouring processes is required when one process
updates its data in the halo area. To facilitate the communications, we initialize three
pairs of arrays: 'send_sites_(v/e/t)' and 'recv_sites_(v/e/t)', for data defined on the
node (v), edge (e) and triangle (t) points, respectively. These arrays are initialized for
each neighbouring process and are used to record the global indices of the data to be
sent to this neighbour as well as the data to be received from this neighbour. Then, the
global indices are converted to the local indices for the ease of data preparations and
assignments.



The inter-process communications are performed by three consecutive steps:
(i) Data preparation. Each process puts the variable data to be sent to the
temporary sending arrays according to the local indices stored in 'send_sites'.
(ii) Data sending and receiving. Data are sent and received using the
non-blocking point-to-point communication interfaces in MPI.
(iii) Data assignment. Each process assigns the received data to the halo area of
this variable according to the local indices stored in 'recv_sites'.
To improve the granularity of data exchange and reduce the number of
inter-process communications, we use a linked list to collect variables that need to be
exchanged. After the collection, the communication interface is called only once to
complete the data exchange of all the variables in the list, which improves the
communication efficiency. When the communications are done, the linked list needs
to be released.
The complicated procedures for communication mentioned above are wrapped
into two subroutines: 'exchange_data_add' and 'exchange_data'. The former one is
used for adding the model variables (whose halo area needs to be updated) to the
linked list. The latter one is used for performing the data exchange and releases the
linked list when the communications are finished. In this way, scientific model
developers only need to decide where and when to utilize these communication tools,
depending on their respective solution techniques and modelling workflow. No
knowledge regarding the details of communication is required, which greatly
facilitates the implementation cost, streamlines the code flow and eases code
refactoring.

**2.3 Scaling tests**

We report the scaling test results to show the efficiency of the partition method
and the communication techniques. All the tests in this paper are carried out on a
Sugon HPC platform. Each computation node contains 64 CPU cores with 256 GB
memory. The Sugon Parastor300 parallel file system is used as the storage system. We
run 60 MPI processes on each node to ensure enough available memory for the tests.





In this paper, we choose the dry hydrostatic dynamical core for testing and
analysis[2]. Two model grids are used: the G10 grid with 10,485,672 grid cells (~7 km
resolution) and the G8 grid with 655,362 grid cells (~30 km resolution). The
timesteps are set to 10 and 40 seconds for G10 and the G8, respectively. Therefore,
the total computational cost of the G10 test is 64x that of the G8 test. The number of
vertical layers is set to 30, and the model integration time is set to 1 day. The results
of the run time with different numbers of processes are shown in Figure 3a. We
choose the run time of G10 simulation with 300 processes as the benchmark, and all
the run times are divided by the benchmark run time. Each run-time point is an
average of three independent runs. The lines of the ideal run time are obtained by
assuming 100% parallel efficiency, which starts from 1 and 1/64 for the G10 grid and
G8 grid, respectively. We may observe that the actual run-time lines are very close to
the ideal run-time lines, suggesting that the model scales well. It should be noted that
all the actual run times of the G10 grid are shorter than the corresponding ideal run
times, that is, the super-linear speedup is achieved for the G10 grid. This abnormal
phenomenon indicates that there is still room for improving the computational
efficiency of running with smaller numbers of processes. For models on the
unstructured meshes, improving the rate of cache hits is an effective way to improve
the computational efficiency. We apply the mesh index reordering strategies for this
purpose. Before entering the next section, Figure 3b first shows the scaling test results
of the BFS index reordering strategy. We can observe that the actual and ideal
run-time lines of the G10 grid are almost coincident. This implies that the index
reordering strategies indeed accelerate the calculations of running with smaller
numbers of processes.

**3 Mesh index reordering strategies**

As is known, the cache is designed to improve the memory-access efficiency of a
CPU. Cache works by improving the data reuse, through which the memory accesses
are replaced by the accesses to the cache. Because the CPU accesses the cache much

---

[2] One may also find in Zhang et al. (2020) (their supplement file), for a strong scaling test that extends from 5120
to 10,240 processes using the moist model with simple physics, a parallel efficiency of ~90% is achieved on a
different machine.





faster than the main memory, the computational efficiency can be improved. Under
the general caching mechanisms, improving the data locality is an efficient way to
enhance the cache reuse. For computing on the unstructured mesh, the stencil
calculations are almost the most computationally intensive tasks. Performing stencil
calculations for a mesh point requires data on its neighbouring points, which is
supported by the indirect addressing scheme. Since the neighbours of a mesh point lie
nearby in the two-dimensional (2D) sphere, it is important to find an indexing strategy
to assign a nearby location in memory for these 2D spatially nearby mesh points.
Generally, the inner area of each process contains most of its mesh points, and
for the application of asynchronous communication technology in the future, we only
reorder the indices of the mesh points in the inner area: it is difficult to apply the
asynchronous communication technology if the mesh points in the inner area and
boundary area are mixed. From the governing equations and the discretization
methods utilized in Zhang et al. (2019), it can be easily deduced that not only the
locality of node points is important but the localities of edge and triangle points are
also important to the cache efficiency. For example, the construction of tangent force
(Thuburn et al. (2009); Ringler et al. (2010)) and the calculation of horizontal flux
(Skamarock and Gassmann (2011); Zhang (2018)) require the loop over the edge
points, while the calculations of Coriolis force and vorticity require the loop over the
triangle points. However, in the practical implementation, only the indices of the node
points need to be reordered. The reason is that the index orders of edge and triangle
points depend on that of the node points, so the locality of node points can ensure the
locality of edge and triangle points.
We apply three index reordering strategies to optimize the locality of the mesh
points: the breadth-first-search (BFS) strategy, the Hilbert curve strategy, and the
Morton curve (a.k.a., Z-order curve) strategy. These indexing strategies help to
generate a distribution of points that has better locality in memory, leading to a higher
cache hit rate and computational efficiency. Before introducing each of them, Figure
4a first shows the mesh index order without reordering. The index order of the node
points is completely chaotic, as the node points are generated by the recursive
bisection of the icosahedral grid with small modifications.

**3.1 The BFS strategy**






The BFS strategy is a graph search algorithm commonly used to solve the
shortest path problem of unweighted graphs, which can be implemented by the
following three steps:
(i) Initialize an empty queue, and select a node point as the first node of the
queue;
(ii) Take out the first node of the queue and then add all its child nodes
(neighbouring points) into the queue (if a child node is already in the queue or has
been in the queue before, it will not be added);
(iii) If the queue is empty, then the procedure ends; otherwise, go to step (ii).
Since the neighbours of each node point are arranged counter-clockwise in the
grid data, the index order of the BFS strategy presents the form as shown in Figure
4b.

**3.2 The Hilbert curve indexing strategy**

The Hilbert curve is a kind of fractal curve, which maps 2D or
higher-dimensional data into one dimensional data and well preserves the spatial
locality. Because the original Hilbert curve indexing strategy is used for regular node
points, we need to convert the unstructured node points into a regular pattern. That is,
the 2D coordinates need to be determined for each node point. This can be
accomplished by establishing an oblique coordinate system, as shown in Figure 5.
First, we need to determine the origin of the system. We choose the first node point
with six neighbours (the hexagon points) in the inner area as the origin, whose
coordinates are (0, 0). After that, the six neighbours of the origin are sequentially
initialized with coordinates +1 or -1 in the x or y directions, that is (0, 1), (-1, 1), (-1,
0), (0, -1), (1, -1), (1, 0) are assigned as the coordinates of the six neighbours in a
counter-clockwise manner. Then, this procedure is repeated for the neighbours'
neighbours until covering all the node points in the inner area. It should be pointed
out that since the non-hexagon points cannot be arranged in the same manner as
hexagon points, special treatment is required when encountering non-hexagon points.
The coordinates of the neighbours of the non-hexagon points are not initialized and
set to the default (0, 0). Since there are only a few non-hexagon points, this has little



impact on the performance.
After the 2D coordinates are initialized, the minimum x and y coordinate values
of all the node points are subtracted from the x and y coordinates, respectively, which
ensures that all coordinate values are non-negative. Since the number of points in the
x and y directions should be $2^n$ (n is a non-negative integer) for the standard Hilbert
curve indexing strategy, we choose the smallest $2^n$ that can cover the largest x and y
coordinate values as the total number of points. Finally, using the x and y coordinate
values of each node point, as well as $2^n$ as the inputs, the standard xy2d function (cf.
https://en.wikipedia.org/wiki/Hilbert_curve) is called to obtain its converted 1D value.
Then, the node points are sorted according to the 1D values, which finishes the
application of the Hilbert index reordering strategy. Figure 4c shows the Hilbert
indexing order in a practical simulation.

**3.3 The Morton curve indexing strategy**

The Morton curve is also a fractal curve analogous to the Hilbert curve. The Morton
curve indexing strategy can be implemented by the following GeoHash algorithm:
(i) Convert the latitudes and longitudes of the node points into binary numbers;
This is done by the bisection method: if a point is in the left sub-interval, we set
0; otherwise, we set 1. Let us take (31, 121) as an example. For the latitude 31, divide
the latitude interval [-90,90] into [-90,0) and [0,90]. Since 31 is in the right interval,
we obtain 1. Then, divide [0, 90] into [0,45) and [45,90]; we obtain 0 as 31 is in the
left interval. Repeat this procedure to obtain the latitude binary number
101011000101110. Then, apply the same strategy to the longitude 121; we obtain the
longitude binary number 110101100101101.
(ii) Merge the binary numbers obtained by step (i);
Put the longitude number on the even digits and the latitude number on the odd
digits. For the case in step (i), we obtain 111001100111100000110011110110.
(iii) Encode the merged numbers according to Base32 and sort the node points
by the encoded strings.
Use the 32 characters (Base32) 0-9 and b-z (remove a, i, l, o) to encode the
merged numbers. Take five consecutive binary digits of a merged number as a group,
which ranges from decimal 0 to 31, and convert it to the corresponding character in





Base32. For example, the merged number in step (ii) is converted to "wtw37q". After
the encoding, we sort the node points according to the character strings to complete
the implementation of the Morton curve indexing strategy. Figure 4d shows the index
order of the Morton curve strategy.

344        Finally, we provide a remark about the relationship between the mesh resolution

and the length of the converted strings. Assume that the length of the string to be
converted is L; then, the total binary digits of the longitude and latitude are 5L. If L is
even, the number of binary conversions for longitude and latitude using the bisection
method is 5L/2; if L is odd, the longitude bisection times is [5L/2]+1, and the latitude
bisection times is [5L/2]. More clearly, the relationship between L and the resolution
is shown in Table 1. Since the target resolution of the densest mesh we currently use is
~3.5 km, setting L=5 is enough to meet our requirements.

**3.4 Numerical tests of the mesh index reordering strategies**

355        In this subsection, we present the performance of the mesh index reordering

strategies through numerical experiments. The model settings are the same as those of
the test cases in subsection 2.3. Three types of grids are used here: the (quasi-uniform)
G10 grid, the quasi-uniform G8 grid, and the variable-resolution G8 grid (a G8X4
gird, which means the fine-mesh and coarse-mesh resolutions vary roughly by a ratio
of 4, and the timestep is set to 20 seconds). The speedups of the index reordering
strategies relative to the original-ordering case with different numbers of processes
and different grids are shown in Figure 6.

363        For the G10 grid, compared with the unoptimized case, the run times of all the

index reordering strategies are reduced, with a speedup ranging from 1.04x to 1.42x.
As the number of processes increases, the optimization effect of using the index
reordering strategies becomes less significant. The reason is that as the number of
processes increases, the number of mesh points on each process decreases, implying
that the percentage of data put into the cache is increased. Therefore, the effect of
cache optimization by using the index reordering strategies becomes less obvious.

370        For the G8 grids, when using the same number of processes with the G10 grid

(see the lower left part of Figure 6), the three index reordering strategies can speed up
the calculations on some test cases, but with a smaller speedup factor. While for the





other test cases, acceleration is relatively hard to achieve. This is because the number
of mesh points distributed to each process is much less than that of the G10 grid. As
we decrease the number of processes, as shown in the lower right part of Figure 6, the
speedups of the three index reordering strategies become conspicuous again. When
running on 60 processes, a 1.12x speedup and a 1.22x speedup are obtained for the
quasi-uniform G8 grid and variable-resolution G8 grid, respectively. These results
suggest that the index reordering strategies can indeed speed up the calculations,
especially for running with a small number of processes.
Based on tests using the three indexing strategies, the BFS strategy typically
performs best and can be used as the default indexing strategy.

**4 The data I/O optimization**

**4.1 The original parallel I/O method**

Except for the communication and computation, the data I/O is an important
issue that may lead to the increase of simulation time, posing a bottleneck for the
high-resolution or massively parallel simulations (see, e.g., Maisonnave et al. (2017);
Koldunov et al. (2019)). This issue becomes especially challenging for the
unstructured-mesh models because of discontinuous accesses. As shown in Figure 7a,
originally, we call the PnetCDF (Li et al. (2003)) interface to perform the I/O
operations, and each process directly interacts with the parallel file system. To give a
more specific example, we use the data input procedure for an illustration. When
reading data in parallel, the global indices of the data to be read by each process are
discontinuous (that is, the positions of the data to be read in the input file are
discontinuous, due to the use of the unstructured mesh), while the interface for
reading data in PnetCDF requires that the data read each time are located
continuously in the input file. Therefore, the reading interface in PnetCDF has to be
called multiple times. To reduce the number of interface calls, we initialize two arrays
'var_start' and 'var_count' to record the starting positions and lengths of the data to be
read by each process, respectively. That is, 'var_start (i)' is the starting position of the
input file for the i-th call to the PnetCDF reading interface, and 'var_count (i)' is the
length of the data for the i-th call to the PnetCDF reading interface. The sizes of these





two arrays are the number of times that the PnetCDF reading interface is called. With these two arrays, we call the PnetCDF nonblocking reading interface 'nfmpi_iget_var' multiple times to read the data. It is worth noting that the data are not imported when calling 'nfmpi_iget_vara', but only the reading requests are recorded. The reading is actually carried out at the wait interface 'nfmpi_wait_all'.

The 'var_start' and 'var_count' arrays are initialized in the mesh partition procedure, and the knowledge of implementation details is not required for scientific model developers. After that, these two arrays can be used as the inputs to call the 'wrap_read_par' function to read the grid data or the variable data. The data output follows the same approach as the data input, except one special treatment: the edge and triangle points are partitioned following the partition of the node points, while each edge or triangle has two or three node points; thus, each edge or triangle point may belong to two or three processes. To avoid the conflicts during the data output, we choose the process with the smallest rank to perform the output of the data defined on the edge or triangle points that belong to more than one process. The users also do not have to know the details of initializing the 'var_start' and 'var_count' arrays for the data output. In addition, similar to the inter-process communications, we have also designed a linked list to collect variables that need to be output. An interface called 'wrap_add_field' can be used to add the variables to the list. When the collection is finished, an interface called 'wrap_output' is used to write all the collected model variables in the list to the parallel file system.

Although the method mentioned above can combine multiple reading requests, PnetCDF shows a significant performance degradation provided that the number of processes scales to several hundreds or thousands. Therefore, we consider improving the I/O efficiency of the parallel infrastructure through the group I/O method.

**4.2 The group I/O method**

As shown in Figure 7b, the processes in the group I/O method are grouped, and only one process in each group (denoted by the I/O process) is responsible for interacting with the parallel file system. The data to be read by other processes are imported through the I/O process and then transmitted from the I/O process through MPI. The data to be output by other processes are sent to the I/O process and then





written to the parallel file system by the I/O process. The group I/O method can improve the I/O granularity by reducing the number of processes interacting with the parallel file system, thus reducing the number of calls to the PnetCDF nonblocking reading/writing interfaces. The group I/O strategy has a much higher efficiency than the original ungrouped parallel I/O and is implemented in several major steps.

The first step to apply the group I/O method is to determine the I/O processes. We use a user-specified parameter 'group_size' to determine the size of the process-groups, i.e., how many processes are in one group. Then, the processes with ranks divisible by 'group_size' are chosen as the I/O processes. For an I/O process with rank i, the processes with ranks ranging from i + 1 to i + group_size − 1 are the non-I/O processes in the same group with process i. Then, as stated in subsection 4.1, the 'var_start' and 'var_count' arrays are initialized for all the processes to record the starting positions and lengths of the data to be input and output. However, for the group I/O method, these arrays are only required for the I/O processes. To initialize the 'var_start' and 'var_count' arrays, the I/O process in each group first gathers the global indices of node, edge and triangle points that distributed to the non-I/O processes, which is accomplished by calling the 'MPI_Gatherv' interface. After that, the I/O process sorts these indices to obtain the largest continuous intervals and builds up maps between the original unsorted and corresponding sorted indices. These maps are used for data rearrangements between the order in the processes and the order in the parallel file system.

Next, the 'var_start' and 'var_count' arrays are determined for the sorted indices of the I/O processes. Then, the group I/O can be carried out when the initialization of 'var_start' and 'var_count' arrays are finished. It should be noted that the communicator for calling the 'open' or 'create' interface in PnetCDF is composed by all the I/O processes, since only the I/O processes interact with the parallel file system. For the data input, the PnetCDF nonblocking reading interface 'nfmpi_iget_vara' and the wait interface 'nfmpi_wait_all' are used as in the original parallel I/O method, but only by the I/O processes. When the reading is done, the I/O process in each group rearranges the data from the sorted-indices order to the unsorted-indices order (the order in the processes) and then calls the 'MPI_Scatterv' interface to send the data to the non-I/O processes. For the data output, the I/O process in each group gathers the data from the non-I/O processes by calling the 'MPI_Gatherv' interface. Then, the I/O





processes rearrange the data from the unsorted-indices order to the sorted-indices
order. Finally, the output is done by the I/O processes through calling the
'nfmpi_iput_vara' and 'nfmpi_wait_all' interfaces.
The complicated operations described above are wrapped by the
'wrap_read_group' and 'wrap_output_group' subroutines for the data input and output,
respectively.

**4.3 Numerical tests**

This subsection examines the performance of the group I/O method. The
(quasi-uniform) G10 grid, the quasi-uniform G8 grid, and the variable-resolution G8
grid (G8X4) are used. The run times of data input and output with different numbers
of processes and 'group_size's are presented in Figure 8. The run times in each
sub-figure are divided by the corresponding run time with 600 processes, and
group_size = 1 (i.e., without grouping).
For the data input, the reading time of the original parallel I/O method
(group_size = 1) increases significantly as the number of processes increases. The
group I/O method with any 'group_size' larger than 1 can reduce the reading time
compared with the original I/O method. For the G10 grid, the best performance is
usually achieved when the number of I/O processes (i.e., the number of processes
divided by the group_size, since there is one I/O process in each group) is near 120,
and more than 90x speedup is observed when the total number of processes is 4200.
For the G8 and G8X4 grids, the best number of I/O processes is between 30 and 70,
and more than 122x speedup and 108x speedup can be achieved for the quasi-uniform
G8 grid and G8X4 grid, respectively, when the total number of processes reaches
497 4200.

For the data output, the group I/O can reduce the writing time for both the G8
and G8X4 grids with almost all the 'group_size's larger than 1, while it is only
effective for the G10 grid with part of the 'group_size's. For the G10 grid, the best
number of I/O processes is between 120 and 200, and more than 3x speedup can be
achieved for all the process numbers. The reason why more speedup is achieved for
the data input than for output may be that the second dimensions of the input data
(smaller than 7, mainly grid data currently) are much smaller than those of the output
data (the number of vertical layers of the variables, 30 in this study). This means that
the input data are 'more discontinuous' than the output data, so the optimization effect
of the group I/O method for data input is more significant than for data output. For the
G8 and G8X4 grids, the best number of I/O processes is between 50 and 80, and more
than 80x speedup and 84x speedup can be obtained for the quasi-uniform G8 grid and
G8X4 grid, respectively, when the total number of processes reaches 4200. These
results demonstrate that the group I/O method can effectively improve the I/O
efficiency of the unstructured-mesh models, especially for the massively parallel
simulations.

**5 Conclusions**

In this paper, we have described the development and performance optimization
of a parallel computing infrastructure for supporting an unstructured-mesh global
model. The work manifests in three aspects, all of which contribute to performance
improvement. The major conclusions are summarized as follows.
(i) The mesh partition accomplished by the METIS library is convenient for both
the quasi-uniform and VR simulations. By designing a general interface with an
effective communication mechanism, scientific model developers only need to decide
where and when to utilize these communication tools, depending on their respective
solution techniques and modelling workflow. No knowledge regarding the details of
communication is required. The scaling tests demonstrate that the partition method
and the communication techniques are efficient.
(ii) The three mesh index reordering strategies are able to improve the
computational efficiency through the cache optimization. The effect is particularly
conspicuous for the high-resolution tests with a relatively small number of processes
(as compared to the total number of cells). The BFS strategy typically performs the
best and is recommended as a default option if index optimization is activated.
(iii) The original parallel I/O method scales poorly due to the discontinuous
feature of the unstructured meshes. To overcome this problem, we have developed a
group I/O method, which can improve the I/O granularity by reducing the number of
processes interacting with the parallel file system. This strategy can significantly
improve the I/O efficiency for massively parallel simulations, especially for global





high-resolution modelling.
The three aspects of the parallel computing toolkits mentioned above are
encapsulated in only a few interfaces that can be used by scientific model developers.
No knowledge regarding the details of parallel implementation is required, thus
reducing the development cost, helping to streamline the code flow and easing the
code refactoring. This approach shares elements of a similar philosophy inherent in
the OpenArray library introduced by Huang et al. (2019), while technically different.
These parallel computing toolkits are not only useful to the existing models but may
also benefit the addition of new dynamical models in the future.
Further, the asynchronous communication technology may be implemented to
overlap the computations of data in the inner area and the inter-process
communications for updating the data in the halo area, which can hide the
communication time and improve the computational efficiency. The heterogeneous
many-core acceleration technique will be applied to port the model to the Sunway
TaihuLight supercomputer for achieving higher computational efficiency.

*Code availability.* GRIST is available at https://github.com/grist-dev, in private
repositories. A version of the model code, running and postprocessing scripts for
supporting this paper are available at: https://zenodo.org/record/3930643. An
authorized link is provided for the editor and reviewers to access the code, which does
not compromise their anonymity. The running scripts are located at:
run_scripts/Perf-test. The grid data used to enable the tests can be downloaded from:
https://zenodo.org/record/3779535. The source code is available to a member of the
model development projects, or people who have interest. Per the current policy on
code sharing at Chinese Academy of Meteorological Sciences, public authorization
may be granted provided that one accepts the terms and conditions:
https://github.com/GRIST-Dev/TermsAndConditions.

*Author contributions.* DW implemented and tested the parallel partition code. ZL
designed the mesh index reordering strategies as well as the group I/O method. Xing
Huang and MW implemented and tested the index reordering strategies and the group
I/O method. Xiaomeng Huang, JL, ZL and YZ led the writing of this paper with
contributions from all other coauthors.




*Competing interests.* The authors declare that they have no conflict of interest.

*Acknowledgements.* This work is supported by a grant from the National Key R&D
Program of China (2017YFC1502203, 2016YFB0201100, 2017YFC1502200,
2018YFB0505000, 2018YFB1502800), the Qingdao National Laboratory for Marine
Science and Technology (QNLM2016ORP0108), the National Natural Science
Foundation of China (41776010), and the Center for High Performance Computing
and System Simulation of the Pilot National Laboratory for Marine Science and
Technology (Qingdao). YZ acknowledges support from the National Natural Science
Foundation of China (41875135) and the National Key R&D Program of China
(2016YFA0602101).

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





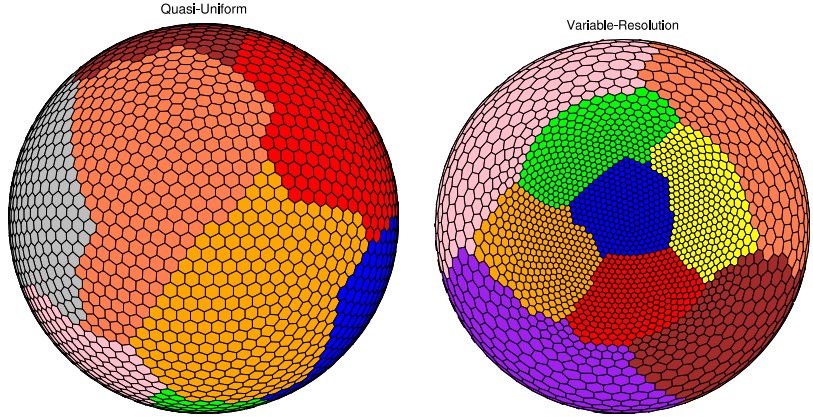

**Figure 1.** The quasi-uniform and VR Voronoi tessellations. Left: the quasi-uniform mesh, Right: the VR mesh. Both meshes are partitioned by METIS, and cells of the same colour belong to the same process.



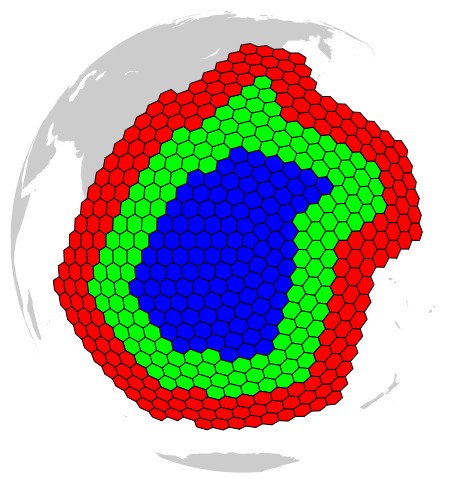

**Figure 2.** The local mesh of one process, consisting of the inner area (blue), the boundary area (green), and the halo area (red), with three layers of halo cells.



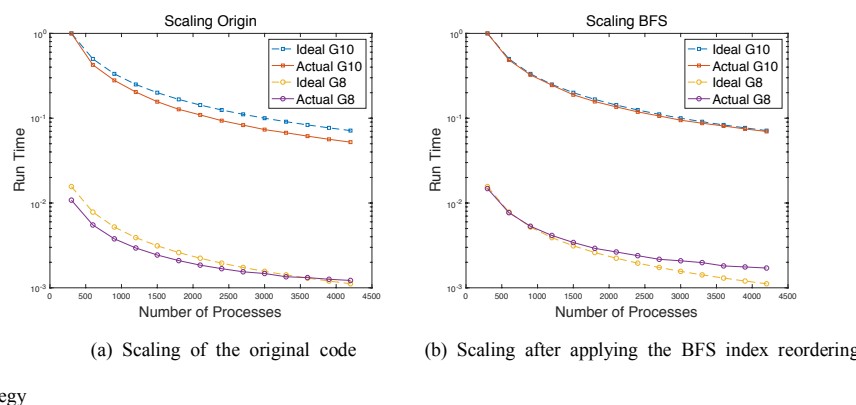

(a) Scaling of the original code   (b) Scaling after applying the BFS index reordering

strategy

**Figure 3.** The ideal and actual run times under different numbers of processes for the G10 and G8 grids. X label: the number of processes, Y label: the total run time (All the run-time points are divided by the corresponding benchmark run time, i.e., divided by the run time of simulation under G10 grid with 300 processes).





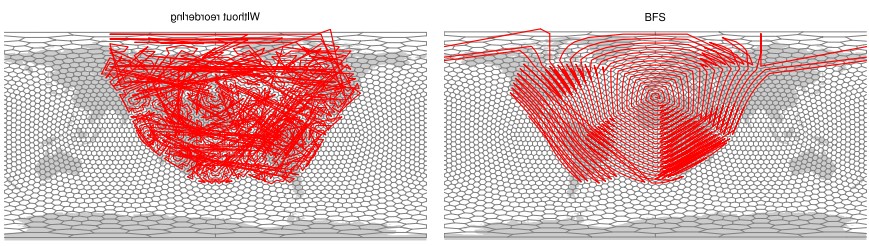

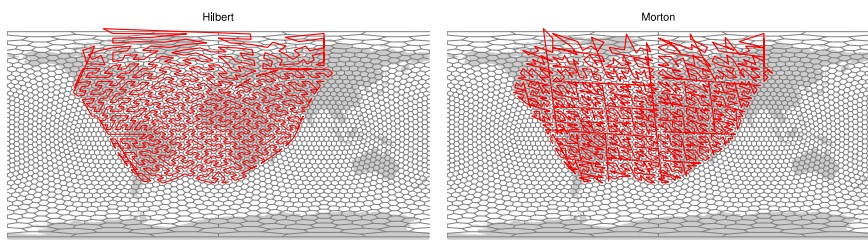

**Figure 4.** The index order of node points in the inner area of process 0 for the G4 grid (2562 node points, running with two processes). Compared with the original-ordering case, the orders of BFS, Hilbert, and Morton strategies appear much better.



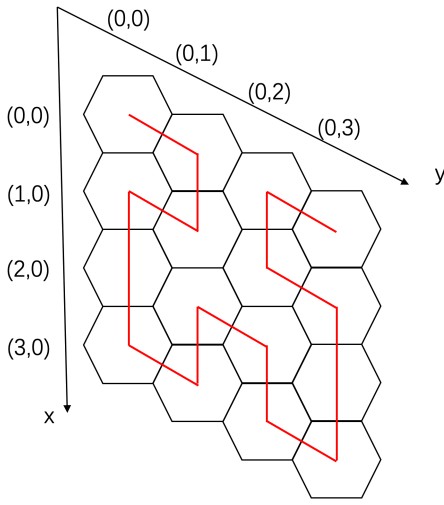

**Figure 5.** The Voronoi polygons and the oblique-coordinate Hilbert curve.





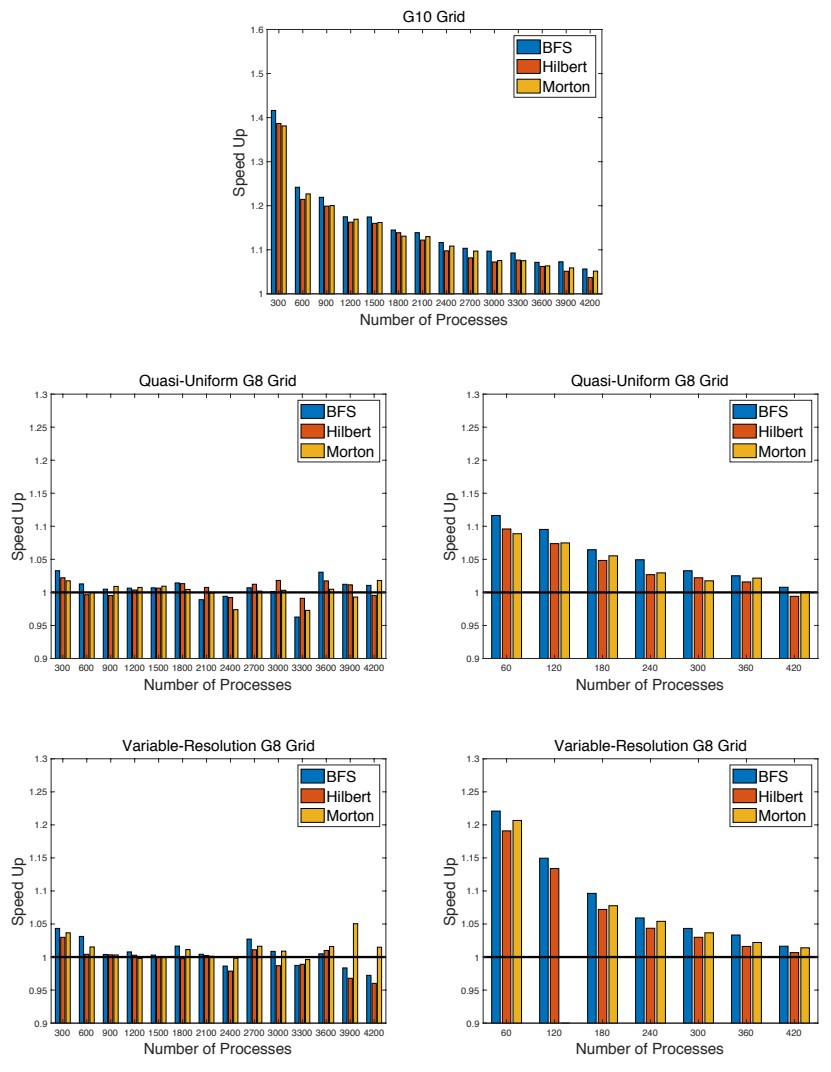

**Figure 6.** The speedups of index reordering strategies relative to the original-ordering case. X label: the number of processes, Y label: the speedup relative to the original-ordering case.





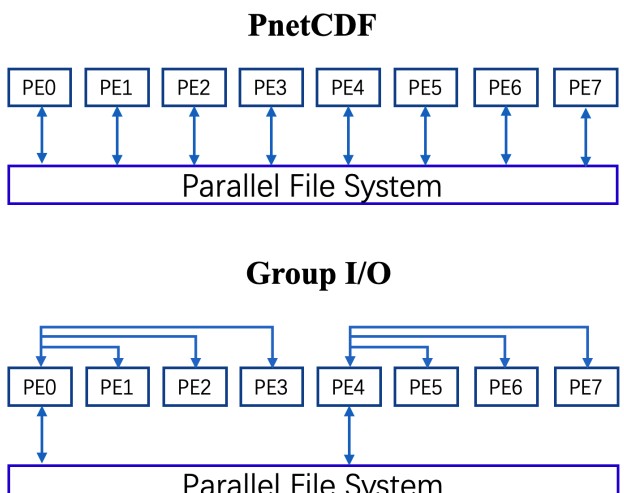

**Figure 7.** The straight PnetCDF I/O method and the group I/O method (group_size = 4). PE (process element) denotes an MPI process.





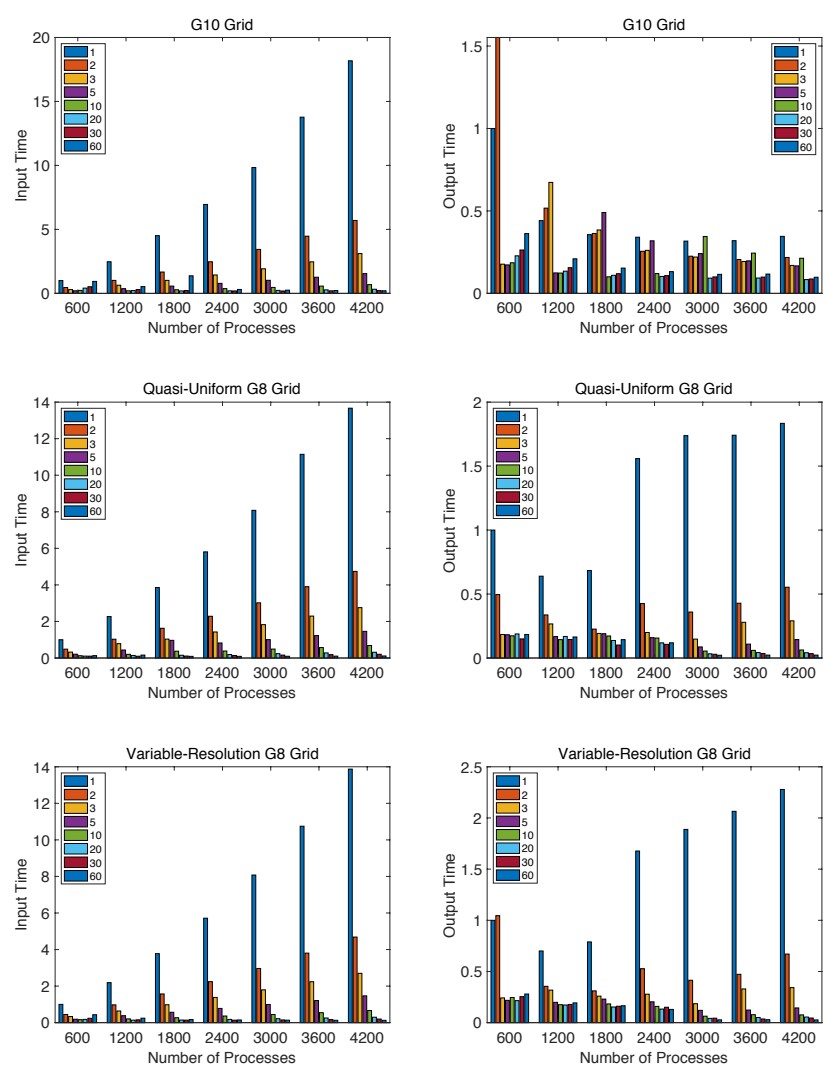

**Figure 8.** The input (left) and output (right) times for different grids. X label: the number of processes, Y label: the run time of data input/output (All the run-time points are divided by the corresponding run time with 600 processes and group_size = 1). Different coloured bars represent results obtained with different 'group_size's.





**Table 1.** The relationship between the mesh resolution and the length of the converted string L.

| L | lat. bisection times | lon. bisection times | lat. resolution (degree) | lon. resolution (degree) | resolution (km) |
|---|---|---|---|---|---|
| 1 | 2 | 3 | 23 | 23 | 2500 |
| 2 | 5 | 5 | 2.8 | 5.6 | 630 |
| 3 | 7 | 8 | 0.7 | 0.7 | 78 |
| 4 | 10 | 10 | 0.087 | 0.18 | 20 |
| 5 | 12 | 13 | 0.022 | 0.022 | 2.4 |
| 6 | 15 | 15 | 0.0027 | 0.0055 | 0.61 |