# Peer review of "Development and performance optimization of a parallel computing infrastructure for an unstructured-mesh modelling framework"

_Geoscientific Model Development, 2020_

## Referee Comment (RC1) · Anonymous Referee #1 · 23 Oct 2020

The paper is well written and clearly describes what work the authors carried out. The explanation of the framework, how it works, and all of the components related to the paper's content are clear and easy to understand. It is also pretty easy to understand from this paper that this framework took a lot of effort and it's important not to underestimate the contribution that a modeling framework provides. The paper explores many avenues for performance improvement, such as examining local mesh reordering techniques to improve cache reuse.

While the paper is well written, it is difficult to understand why some of the choices were made. The paper heavily references another modeling framework, MPAS, which

does largely the same thing as GRIST. The overarching discretization, mesh choices, decomposition methods, and parallel I/O strategies seem almost identical to those used in MPAS. It would be good to have some additional paragraphs in the paper describing why this new modeling framework is an addition to the community on top of the existing open-sourced modeling frameworks. Two suggestions for this would be:

A new paragraph in the introduction explaining why this framework is necessary given other frameworks that already exist in an open-sourced format. I think this would help avoid any assumptions of redundancy that could be confusing to readers familiar with other efforts in this space. A new paragraph in the conclusion comparing and contrasting this new framework with other existing frameworks. This could also just be a table or whatever format is easy for the authors.

Aside from that, the paper positions itself as if it is going to look at many sweeping avenues of performance optimization, and while something like local mesh reordering is potentially important (especially if the users primarily focus on low processor count runs) it can easily be out performed by studying the on-node performance impacts, or even looking at inefficiencies from the MPI implementation. In future work (because it is a lot of effort, and probably outside of the scope of this paper) I'd highly recommend looking at how well the model vectorizes, and performs on a single node before looking at things like cache reuse. Especially since the data provided (and previous studies on this subject) show that you can negate the impact of reordering by strong scaling your model out further. This also means you could simply over-decompose your problem to avoid having to reorder data. While the authors argue that their framework shows a good scaling efficiency, it's hard to accurately estimate how good the strong scaling effects of the framework are without some serial baseline numbers. As a result, I think it would be beneficial for the authors to try and add some serial (and lower processor count) data to figure 3. It looks like the authors stop exploring their strong scaling space at around 250 processes, and it's hard to judge what the loss in parallel efficiency is without these numbers. In the end, I would suggest three major revisions elements,
mainly to help emphasize the contributions that this framework brings to the community, and to help showcase how efficient the framework is as a whole. These are listed below: A new introductory paragraph as mentioned above A new concluding paragraph or table as mentioned above Adding lower processor data to the strong scaling plots. Ideally all the way down to a single process.

Outside of those major revision elements, here are some minor points: While the paper as a whole recognizes prior contributions by frameworks such as MPAS, the abstract does not. It would be useful to add something to that effect into the abstract. Based on the usage of METIS, I'm assuming that the authors use the offline capability (specifically in METS) and not the online capability in ParMETIS, but this could be clarified Line 40: "which reduces the" -> "which can reduce the". Line 80: "establishment of numerical modelling" -> "establishment of a numerical modelling" Section 2.3: You never mention that the G8 performance actually is degraded by using the BFS reordering strategy. This is also mentioned again on lines 370-372. This is important to notice, because the reordering strategies become unnecessary once the problems are strong scaled out to a certain point. It could also be useful to give a description of what this point is as a function of number of cells / process. Section 4.2: This essentially describes NCAR's PIO library, which is available on github. Though no reference to it appears in this section, and no discussion of why this was rewritten instead of used wholesale appears. This should definitely be added. It could be as simple as describing why it is not the same as NCAR's PIO library, and moving on from there though. Figure 4: The title on the top left sub-plot is mirrored

Referee criteria: Scientific significance: 3-4 (Fair - Poor) Scientific Quality: 2 (Good) Scientific Reproducibility: 2 (Good) Presentation Quality: 1 (Excellent)
* * *

---

## Referee Comment (RC2) · Anonymous Referee #2 · 27 Oct 2020

This paper describes the aspects of parallel computing, index optimization and an efficient group I/O strategy in the development of a parallel computing infrastructure for an unstructured-mesh global model. Computational performance of an unstructured-mesh global model is an important research topic as it impacts overall model performance especially in the high resolution grids. This paper is composed of the detailed steps to construct mesh using METIS tool, to communicate between processes using MPI non-blocking APIs, to evaluate three mesh index reordering strategies, to select a process for group communication, and to use MPI API for completing the group communication. Finally, numerical test results are provided with varying process numbers showing the speed-ups. Overall, this paper has merit of exploring multiple options in its

design and implementation with enough details. However, it is not clearly addressed claiming the parts that are noble in this paper.

Scientific questions / issues

1. According to several parts of this paper, three mesh index reordering strategies are investigated to improve cache efficiency. However, there is no discussion about the cache efficiency differences in the three index reordering strategies. Therefore it is unclear if the outcome of the investigation is due to cache efficiency or something else.

2. In section 2.2, it is claimed that scientific model developers can implement communication without knowing the communication details by using "exchange_data_add" and "exchange_data" based on "linked list". It is unclear how much details of communication this tool hides when compared to conventional methods.

3. It is unclear if the timing of group I/O in Fig. 8 includes any overhead for implementing group I/O such as selecting one designated process for communication.

4. In section 2.3, there are statements: "It should be noted that all the actual run times of the G10 grid are shorter than the corresponding ideal run times, that is, the super-linear speedup is achieved for the G10 grid. This abnormal phenomenon indicates that there is still room for improving the computational efficiency of running with smaller numbers of processes." It is unclear why the super-linear speedup happens. Also, it is also unclear why the super-linear speedup indicates that the room for improving the computational efficiency exists.

5. In the paper, there are terms of "toolkit" and "framework". It is unclear what is the relationship between them.

---

## Short Comment (SC1) · 27 Oct 2020

Dear authors,

in my role as Executive editor of GMD, I would like to bring to your attention our Editorial version 1.2:

https://www.geosci-model-dev.net/12/2215/2019/

This highlights some requirements of papers published in GMD, which is also available on the GMD website in the 'Manuscript Types' section:

http://www.geoscientific-model-development.net/submission/manuscript_types.html

[Figure]

In particular, please note that for your paper, the following requirement has not been met in the Discussions paper:

- "The main paper must give the model name and version number (or other unique identifier) in the title."

Please add the acronym (GRIST) and a version number in the title upon your revised submission to GMD.

Yours,

Astrid Kerkwe
* * *

---

## Author Comment (AC1) · 1 Dec 2020

**Dear editor and reviewer,**

  **First of all, we would like to express our sincere appreciation to your valuable feedbacks. Your comments are highly insightful and enable us to substantially improve the quality of our manuscript. Below are our point-by -point responses to all the comments. The red fonts describe how the manuscript has been modified.**

**Responses to the comments of referee #1**

**Major revision comments**

1. Major revision comments 1 and 2. "A new paragraph in the introduction explaining why this framework is necessary given other frameworks that already exist in an open-sourced format. I think this would help avoid any assumptions of redundancy that could be confusing to readers familiar with other efforts in this space. A new paragraph in the conclusion comparing and contrasting this new framework with other existing frameworks. This could also just be a table or whatever format is easy for the authors."

**[Response]:**

To address these concerns and to maintain the overall structure of this manuscript, we have added a discussion section. It now clearly explains why this framework is necessary to us, and its broad implication to the community. For the comparison with existing frameworks, while GRIST implements some techniques and choices that have been already used in the community, its scientific models are different. Developing these models is closely connected

with the parallel infrastructure, requiring tailored software engineering efforts. We provided Table 2 to give some details of GRIST. This table follows the conventions of Ullrich et al. (2017) and briefly summarizes some key features. We make a comparison with three models in Ullrich et al. (2017) to describe the unique aspects of GRIST. All this information is given in Section 5.

These two paragraphs in Section 5 explain why this framework is necessary for us: "First, the authors understand…", "Second, our intention for global atmospheric modelling…".

This paragraph in Section 5 discusses the broad implication: "While GRIST is still under active development…"

This paragraph in Section 5 compares GRIST with three existing counterparts that have been used for weather and climate modelling (MPAS, ICON, FV3): "As a response to one reviewer, it is also worthwhile to pinpoint how GRIST differs from existing counterparts…". Table 2 can be compared to the tables in Ullrich et al. (2017).

2. Major revision comment 3. "I think it would be beneficial for the authors to try and add some serial (and lower processor count) data to figure 3. It looks like the authors stop exploring their strong scaling space at around 250 processes, and it's hard to judge what the loss in parallel efficiency is without these numbers."

**[Response]:**

We have added the test results with lower processor counts in Section 2.3, including two subfigures in Figure 3 (Figures 3(c) and 3(d), also illustrated below) and a new paragraph discussing the results. Due to the memory limitation, we are not able to run the tests with

processor counts lower than 300 for the G10 grid. Therefore, in Figures 3(c) and 3(d), we have shown the results for the G8 grid with processor counts ranging from 2 to 4200, from which we can observe the super-linear speedup phenomenon for low processor counts, and the index reordering strategies can accelerate the calculations. The parallelized code requires at least two processors to run. The one-to-many consistency is guaranteed by maintaining a separate serial version that only contains a minimum code segment for examinations, but this serial version is not very suitable for a reference of computational performance.

[Figure]

Figure 3 (c)

[Figure]

Figure 3 (d)

**Minor points**

1. Minor point 1: While the paper as a whole recognizes prior contributions by frameworks such as MPAS, the abstract does not. It would be useful to add something to that effect into the abstract.

**[Response]:**

We have improved the abstract to address this concern.

"GRIST is a new framework in the icosahedral-/Voronoi-mesh modelling community for

both research and application purposes. It adopts some well-established techniques and choices that have been used, but supports different scientific models. Developing these models is closely connected with the parallel infrastructure, requiring tailored software engineering efforts. In this paper, we focus on three major aspects that facilitate rapid iterative development".

2. Minor point 2: Based on the usage of METIS, I'm assuming that the authors use the offline capability (specifically in METS) and not the online capability in ParMETIS, but this could be clarified.

**[Response]:**

Yes, we are using the offline capability of METIS. This has been clarified in Section 2.1 in the revised manuscript. For GRIST, this METIS-offline partition can be done either online (in the initialization) or offline (via a separate driver), and the offline setup is more suitable for very high-resolution runs, since the mesh partition will consume more time than simply reading the partitioned data for high-resolution simulations.

3. Minor point 3: Line 40: "which reduces the" -> "which can reduce the". Line 80: "establishment of numerical modelling" -> "establishment of a numerical modelling"

**[Response]:**

Thanks. We have corrected these statements.

4. Minor point 4: Section 2.3: You never mention that the G8 performance actually is degraded by using the BFS reordering strategy. This is also mentioned again on lines 370-372. This is important to notice, because the reordering strategies become unnecessary once the problems are strong scaled out to a certain point. It could also be useful to give a description of what this point is as a function of number of cells / process.

**[Response]:**

We have added the statement of the degradation phenomenon of BFS strategy for the G8 performance in the last paragraph of Section 2.3, and referred to Section 3.4 for the discussions. In the last paragraph of Section 3.4, we have added the sentence "Based on our tests, we find out that the index reordering will become unnecessary when there are less than 1000 cells/process". Thanks for your comments.

5. Minor point 5: Section 4.2: This essentially describes NCAR's PIO library, which is available on github. Though no reference to it appears in this section, and no discussion of why this was rewritten instead of used wholesale appears. This should definitely be added. It could be as simple as describing why it is not the same as NCAR's PIO library, and moving on from there though.

**[Response]:**

We have added the discussion with the PIO library in the second paragraph of Section 4.2. PIO is a high-level parallel I/O library for the structured grid applications, which also allows to designate some subset of processors to perform I/O like the group I/O method introduced in this manuscript. We note that MPAS uses PIO as its tool. When compared with PIO, the communications of data between the non-I/O processes and the I/O processes for the group I/O method are much easier. This is because the communications for the group I/O method are only carried out between processes in the same group, which can be accomplished by using the 'MPI_Scatterv/MPI_Gatherv' interfaces, and the indices in the I/O processes are not required to be continuous. While for PIO, the indices in the I/O processes should be continuous, therefore the communications between the non-I/O processes and the I/O processes will be more complicated, and the 'MPI_Alltoallv' interface may be used. Anyway,

the group I/O method is more tailored for the data structure and distribution of GRIST, and thus can be implemented more easily.

6. Minor point 6: Figure 4: The title on the top left sub-plot is mirrored.

**[Response]:**

Thanks. We have corrected this.

We really appreciate your highly constructive comments. If there are any other questions, please do not hesitate to contact us.

Best wishes,

Xiaomeng Huang

---

## Author Comment (AC2) · 1 Dec 2020

**Dear editor and reviewer,**

First of all, we would like to express our sincere appreciation to your valuable feedbacks. Your comments are highly insightful and enable us to substantially improve the quality of our manuscript. Below are our point-by -point responses to all the comments. The red fonts describe how the manuscript has been modified.

**Responses to the comments of referee #2**

1. According to several parts of this paper, three mesh index reordering strategies are investigated to improve cache efficiency. However, there is no discussion about the cache efficiency differences in the three index reordering strategies. Therefore, it is unclear if the outcome of the investigation is due to cache efficiency or something else.

**[Response]:**

We have improved the presentation related to this part in the third paragraph of Section 2.3 and the last paragraph of Section 3.4. The only difference between the reordered and the original codes are the loop ordering, which mainly affects the data locality during model iterations. As is known, the data locality is an important factor to the cache reuse, which has a strong effect on the efficiency of memory access. This is the reason why the index reordering strategies work. Similar speedup results from index reordering methods are obtained in Sarje et al. (2015), where two space filling curve (SFC) index reordering strategies (Hilbert and Morton curves) are used and obtained 40% improvement. Therefore,

we think it is sufficient to conclude that the speedup is due to the cache efficiency. Note that the three index reordering strategies behaves much similar, all of which are able to accelerate the calculations with a similar amount of time. Therefore, we did not make further discussions about the differences in the three strategies.

2. In section 2.2, it is claimed that scientific model developers can implement communication without knowing the communication details by using "exchange_data_add" and "exchange_data" based on "linked list". It is unclear how much details of communication this tool hides when compared to conventional methods.

**[Response]:**

In "exchange_data", we packed the conventional communication interfaces (MPI nonblocking communication interfaces) and made some data assignment before and after the communication, which is explained in the three steps (i), (ii), and (iii) of Section 2.2. The "linked list" is used to enable communicating multiple variables (variables that added to this list) at the same time. The interface "exchange_data_add" is used to add variables whose Halo area need to be updated to this "linked list", as stated in Section 2.2. The aim of packing is to simplify the communication procedure and ease the model development.

We have provided an example in the end of Section 2.2 to more clearly demonstrate the use of these tools given specific solution methods and workflow.

3. It is unclear if the timing of group I/O in Fig. 8 includes any overhead for implementing group I/O such as selecting one designated process for communication.

**[Response]:**

The timing in Fig. 8 does not include the overhead for implementing the group I/O method. The additional work for using the group I/O method is done only once in the model initialization part, which consumes much less time than performing the I/O during model integration. Therefore, the overhead for implementing the group I/O method is negligible and not shown in this figure. We have clarified this point in the first paragraph of Section 4.3 in the revised manuscript.

4. In section 2.3, there are statements: "It should be noted that all the actual run times of the G10 grid are shorter than the corresponding ideal run times, that is, the superlinear speedup is achieved for the G10 grid. This abnormal phenomenon indicates that there is still room for improving the computational efficiency of running with smaller numbers of processes." It is unclear why the super-linear speedup happens. Also, it is also unclear why the super-linear speedup indicates that the room for improving the computational efficiency exists.

**[Response]:**

In the third paragraph of Section 2.3, we have added the following sentences to explain the super-linear speedup phenomenon:

"This abnormal phenomenon arises mainly because of the inefficiency of running with low processor counts. Based on our analysis and numerical experiments, this inefficiency results from the less cache hits rate due to the discontinuous memory access of an unstructured-grid model, which has a much stronger impact on running with low-processor counts than running with high-processor counts. The reason is that as the number of processors increases, the number of mesh points distributed to each processor decreases, implying that the

percentage of data that can be put into the cache is increased and therefore the cache hits rate is increased. This leads to the super-linear speedup phenomenon."

Based on previous studies on the unstructured-grid models (e.g. Sarje et al. 2015), improving the data locality during model integration is an effective way to improve the cache efficiency, especially for running with low processor counts. This is the reason why we implemented the mesh index reordering strategies. We have improved the presentation related to this part in the fourth paragraph of Section 2.3.

5. In the paper, there are terms of "toolkit" and "framework". It is unclear what is the relationship between them.

**[Response]:**

In our terms, "modelling framework" denotes the entire modelling system, including the parallel infrastructure and the scientific models. "Toolkit" denotes several key tools (which is the major focus of this study) offered by the parallel infrastructure to the scientific models. The parallel infrastructure also has other necessary functions. They are not mentioned in this paper because they are either more scientifically oriented (e.g., computation of mesh weight), or too common to be specifically described (e.g., time management, error handling). We have added necessary information in the introduction (see footnote 2) to more clearly explain these terms.

We really appreciate your highly constructive comments. If there are any other questions, please do not hesitate to contact us.

Best wishes,

Xiaomeng Huang

---

## Author Comment (AC3) · 1 Dec 2020

Dear editor,

Thanks for your comments, we have added the acronym and version number in the title. Now the title is "Development and performance optimization of a parallel computing infrastructure for an unstructured-mesh modelling framework (GRIST-A20.7)".

Best wishes, Xiaomeng Huang